# PROVABLY NOISE-RESILIENT TRAINING OF PARAMETERIZED QUANTUM CIRCUITS

## ABSTRACT

Advancements in quantum computing have spurred significant interest in harnessing its potential for speedups over classical systems. However, noise remains a major obstacle to achieving reliable quantum algorithms. In this work, we present a provably noise-resilient training theory and algorithm to enhance the robustness of parameterized quantum circuits. Our method, with a natural connection to Evolutionary Strategies, guarantees resilience to parameter noise with minimal adjustments to commonly used optimization algorithms. Our approach is function-agnostic and adaptable to various quantum circuits, successfully demonstrated in quantum phase classification and quantum state preparation tasks. By developing provably guaranteed learning theory with quantum circuits, our work opens new avenues for practical, robust applications of near-term quantum computers.

## 1 INTRODUCTION

In the past few decades, the field of quantum computing has grown dramatically (Farhi et al., 2014; Harrigan et al., 2021; Moll et al., 2017; Klco et al., 2018; Peruzzo et al., 2014). Enticed by the potential of speed-ups over classical computers, great efforts have been devoted to not only building quantum computers (Bluvstein et al., 2024), but also exploring how to achieve these speed-ups when the proper devices exists (Biamonte et al., 2017) with a wide variety of approaches, from carefully-crafted algorithms (Dalzell et al., 2023) to data-learned optimized approaches (Cerezo et al., 2021).

However, all existing quantum algorithms suffer from the existence of noise on the current quantum hardware. Until the theory and practice of error-corrected quantum computing makes significant progress, all quantum algorithms need to find ways to be robust to the noise that exists in noisy intermediate-scale quantum (NISQ) computers (Preskill, 2018). Furthermore, there are many types of noise, from noise inherent to quantum systems that can't be fully isolated from their environment, to simple instrumentation noise that occurs whenever preparing a continuously-parameterized operation (Saki et al., 2021; Cai et al., 2023). As a result, a number of works have explored different ways to fight noise in the NISQ era (West et al., 2023; Weber et al., 2021; 2022).

In this work, we offer another tool to fight against noise for algorithms deployed on NISQ devices. Specifically we tackle parameter gate noise, in which the prepared parameters of a continuously-parameterized quantum gate may be different than it is intended to be. Our method is a *certified robustness* method, in which we can guarantee a certain level of noise will not affect the end result of the quantum process. We highlight our contributions as follows:

1. Our work provides a provably guaranteed framework and theory for training parameterized quantum circuit classifier under parameter noise. While most other methods consider adversarial attacks on inputs or mid-circuit noise inherent to quantum systems, we explicitly tackle the use case that accounts for instrumentation error in devices. As long as the desired noise-model and algorithm can be expressed as a parameterized circuit, our method can be used to find an optimal robustness certificate for changes in those parameters.

2. Our method is simple, easy to deploy, and naturally connected to Evolutionary Strategies (ES), an optimization algorithm already commonly used by the variational quantum algorithm (VQA) community (Gil-Fuster et al., 2023; Anand et al., 2021). Due to its ability to avoid computing costly quantum gradients (Abbas et al., 2023), a practitioner can achieve robustness certificates using our method with minimal effort. Furthermore, our method

can be applied on top of any error-mitigation method, allowing to be easily combined with other methods to further boost the robustness they provide.

3. Our approach is successfully demonstrated on quantum phase classification and quantum state preparation tasks. Our results present clear robustness-variance trade-off and different robustness-variance correlation in different tasks, which provide insights and understanding on the sensitivity of the parameters in quantum parameterized circuits.

In summary, we develop a provably noise-resilient training theory and algorithm for parameterized quantum circuits. With its flexibility and ease of adaptation, our approach achieves noise robustness, paving the way for practical applications on near-term quantum computers.

## 2 PRIOR WORK

**Certified Robustness :** In classical machine learning, there is a large body of work that aims to make models more robust to attacks (Madry et al., 2017; Silva & Najafirad, 2020; Zhang et al., 2019; Pitropakis et al., 2019). A sub-section of this work is *certified* robustness, where bounds and theorems are derived to allow one to prove their model is robust to attacks. One large portion of this work focuses on exactly computing the possible output space of a model given a defined input space, which is usually defined to be the desired input plus the possible space of attack-perturbations. Often this requires incorporating relaxations that make the computed space not strictly tight, but allow the bound to be computed in non-exponential time. Initially these relaxations were devised to accommodate for common non-linear activations in neural networks (Wong & Kolter, 2017; Zhang et al., 2018), but these works have since been extended to allow for more general forms of relaxations and can be formulated into semi-definite programming (SDP) problems (Raghunathan et al., 2018; Salman et al., 2020c).

Another method used to certify robustness is randomized smoothing. In essence, by instead certifying a "smoothed" classifier, which predicts the most likely class your underlying model will output if your input has noise added to it, one can achieve robustness guarantees to perturbations on said input. This was first applied to achieve robustness guarantees for L2 norm perturbations on neural networks, and was even shown to be provable optimal for any given smoothed classifier (Cohen et al., 2019). There has since been numerous extensions. Some extensions allow for extra flexibility in the method, allowing for the use of different noise models, different optimal-robustness guarantees (Yang et al., 2020), different certified regions (Levine & Feizi, 2019; Dvijotham et al., 2019), or more flexibility in customizing the amount of noise per input parameter (Tecot & Hsieh, 2021). Other extensions focus on using various new training techniques to produce better smoothed classifiers (Salman et al., 2020b; Zhang et al.), or devise ways to achieve good smoothed models using a pre-trained model as a base (Salman et al., 2020a).

**Quantum Certified Robustness :** Although a newer area, there has been a sizable amount of interest in studying adversarial robusness in the quantum setting (West et al., 2023; Lu et al., 2020). Just like with classical robustness, these works typically either use SDP to calculate possible output spaces, or use randomized smoothing as a tool to get robustness bounds. On the SDP side, tools and methods have been developed to apply similar tactics in the classical setting but for analyzing quantum models. They have been tested in on a variety tasks for a variety of noise models (Weber et al., 2022; Guan et al., 2021; Lin et al., 2024). Similarly, a variety of works have developed quantum-versions of randomized smoothing, usually either by adding noise to the input and applying randomized smoothing theorems (Du et al., 2021; Huang et al., 2023) or by developing quantum-versions of the smoothing process (Gong et al., 2024; Sahdev & Kumar, 2022). However, Weber et al. (2021) takes arguably a more fundamental approach by realizing that because measuring quantum states is already a probabilistic distribution, one can treat repeatedly measuring a quantum state itself as a randomized smoothed model. From this you can derive bounds relating to the measured state, and extrapolate this to different bounds on noise that may occur.

**Evolutionary Strategies :** Evolutionary Strategies (ES) are a group of algorithms that take a population, sample it for a desired property, and then change said population to on average improve on said property. At least in the context of using this method for continuous optimization, often the population is modelled as a probability distribution, and to optimize it gradient descent is performed on the parameters of the distribution. There are numerous different variations of ES (Li

et al., 2020; Maheswaranathan et al., 2019), but among the most common are Covariance Matrix Adaptation Evolution Strategies (CMA-ES) (Hansen, 2006) and Natural Evolution Strategies (NES) (Wierstra et al., 2011). At their core, both of these methods simply take a multivariate Gaussian, sample it, evaluate the loss function at each sample, and perform gradient descent on the parameters of the Gaussian from said samples. They then repeat this process until convergence. They each have their own ways to improve both the practicality and theoretical guarantees of this process, however it has been show that these two methods are just re-formulations of the same process and are extremely similar (Akimoto et al., 2010). Furthermore, both of these methods are commonly used for optimizing parameterized quantum circuits (Gil-Fuster et al., 2023; Anand et al., 2021).

## 3 METHOD

### 3.1 NOISE-RESILIENT THEOREM

**Definition 3.1 (Parameterized Quantum Circuit (PQC) Classifier)** *A classifier $C$ is called a parameterized quantum circuit (PQC) classifier if it is constructed in the following way:*

$$C(\theta, x)_i = \langle \phi_0 | V^\dagger(x) U^\dagger(\theta) A_i U(\theta) V(x) | \phi_0 \rangle$$
$$U(\theta) = U_L(\theta_L) \cdots U_2(\theta_2) U_1(\theta_1)$$
$$U_l(\theta_l) = \prod_m e^{-i\theta_m H_m} W_m$$

where $C(\theta, x)_i$ is the quantum classifier's probability assigned to class $i$, $A_i$ are easily measurable observables, $V(x)$ is a data-dependent unitary, $W_m$ is an unparameterized unitary, $H_m$ is a Hermitian operator, and $\theta_l$ is the $l$-th element of $\theta$. PQC classifiers have been considered in various quantum machine learning setups (Cerezo et al., 2021; Schuld et al., 2020; Weber et al., 2021). While we follow the above particular form of PQC in this work, our discussion is general as long as the classifier comes from a quantum circuit with unitaries depending on $x$ and $\theta$.

In this work, we focus on noise effect on the given PQC classifier parameters. If the PQC classifier parameters are fully robust to noise, any possible noisy perturbation on the parameters should not change the correct classification of the task. More precisely, we formalize this notion as follows.

**Definition 3.2 (Parameter Noise-resilient PQC Classifier)** *A PQC classifier is parameter noise-resilient if the following is true*

$$\forall (x, y) \in D, \left( \arg\max_i C(\theta, x)_i = y \right) \implies \left( \arg\max_i C(\theta + \delta, x)_i = y \right)$$

for any noisy perturbation $\delta$ from a certain distribution.

Our goal is to develop robust training theory and algorithms for PQC classifier so that it is provably parameter noise-resilient. To tackle this problem, we integrate approach from classical machine learning with parameterized quantum circuits. More specifically, we develop randomized smoothing certified robustness theory (Tecot & Hsieh, 2021) under the setting of PQC classifier. To start with, we introduce the concept of smoothed PQC classifier.

**Definition 3.3 (Smoothed PQC Classifier)** *Let $C$ be a PQC classifier with possible prediction classes $\gamma = \{1, \ldots, N\}$. A smoothed PQC classifier $G_\sigma$ is defined as:*

$$G_\sigma(\theta, x) = \arg\max_{i \in \gamma} \mathbb{P}(C(\theta + \epsilon, x)_i),$$

*where $\epsilon \sim \mathcal{N}(0, \Sigma)$ and $\Sigma$ is a diagonal matrix with vector $\sigma^2$ as the diagonal.*

**Theorem 3.1 (Noise-resilient Condition for Smoothed PQC Classifier)** *Let $C$ be a PQC classifier, $G_\sigma$ to be the corresponding smoothed PQC classifier and $c_a$ be the class for $x$, then $G_\sigma(\theta + \delta, x) = c_a$ for any $\delta$ vectors that satisfy*

$$\|\delta \oslash \sigma\|_2 < \frac{1}{2} \left( \Phi^{-1}(p_A) - \Phi^{-1}(p_B) \right)$$

*where $\oslash$ is the Hadamard (element-wise) division, $\| \cdot \|_2$ is the $L_2$ norm, $\Phi^{-1}$ is the inverse of the standard Gaussian CDF, and*

$$p_A = \mathbb{P}\left( \arg\max_{i \in \gamma} C(\theta + \epsilon, x)_i = c_a \right)$$

$$p_B = \max_{c \neq c_a} \mathbb{P}\left( \arg\max_{i \in \gamma} C(\theta + \epsilon, x)_i = c \right)$$

*with $\epsilon \sim \mathcal{N}(0, \Sigma)$ and $\Sigma$ is a diagonal matrix with vector $\sigma^2$ as the diagonal.*

Using the above theorem, by smoothing a given PQC (i.e. re-evaluating the model multiple times, with $\epsilon$ sampled from a multivariate Gaussian each time), we can guarantee with high probability that any noise $\delta$ caused by the environment will not change the prediction results, as long as it satisfies the given bound. All we need to do is estimate bounds on $p_A$ and $p_B$ using concentration inequalities and then we can directly apply the theorem (See Section B.3 for more details). Since standard PQC classifiers require multiple evaluations due to the probabilistic nature of quantum measurements, our smoothed PQC classifier operates in practice similarly to a standard PQC classifier in both method and computational cost. The implementation only involves determining a $\sigma$ vector in addition to the ideal $\theta$ parameters for sampling, making it resource-efficient for near-term quantum computers. We further note that the ability of our approach to vary $\sigma$ across different parameters, as opposed to using a uniform robust radius, adds flexibility to enhance the system's resilience to noise.

### 3.2 TRAINING FOR ROBUSTNESS

Theorem. 3.1 has provided a provably guarantee on noise resilience for PQC classifier. Next, we consider how to train our circuits and improve this robust-bound. First, let us consider a commonly used method for training PQCs - Evolution Strategies (ES). This optimization algorithm is commonly used in the quantum community (Gil-Fuster et al., 2023; Anand et al., 2021) due to its ability to optimize while avoiding computing costly quantum gradients (Abbas et al., 2023). In general, what ES typically optimizes for is

$$J = \mathbb{E}_{(x,y) \in D, \epsilon \sim \mathcal{N}(0,\Sigma)} O(\theta + \epsilon, x, y)$$

where $D$ is our training dataset, and $O$ is the objective function. While the "search distribution" that we sample $\epsilon$ from can vary depending on the version of ES, we consider a multivariate Gaussian which is the distribution commonly used by the most popular versions of ES in PQC optimization, such as CMA-ES and NES (Hansen, 2006; Wierstra et al., 2011).

**Connection of ES and Noise-resilient Condition.** We highlight that the equation of ES is closely related to the right-side of the bound in Theorem 3.1. In ES, we optimize the parameters of a multivariate Gaussian to minimize an objective in expectation. For Theorem 3.1, we desire to optimize $\theta$ and $\sigma$ to maximize our robust bound. Note that $\theta + \epsilon$ is also a multivariate Gaussian, where $\theta$ is the mean and $\sigma$ are the independent variances. Therefore, if we simply change the objective function $O$ to calculate the margin of prediction instead, we can exactly optimize for the right-hand side of Theorem 3.1 using ES. More precisely, we can re-formulate the optimization goal as:

$$\arg\max_{\theta,\sigma} \left[ \mathbb{E}_{(x,y) \in D, \epsilon \sim \mathcal{N}(0,\Sigma)} O(\theta + \epsilon, x, y) \right]$$

$$O(\theta + \epsilon, x, y) = \frac{1}{2}\left( \Phi^{-1}(p_A) - \Phi^{-1}(p_B) \right)$$

where $\Sigma, p_A, p_B$ are all as defined in Theorem 3.1. Since our theorem relies on having independent variances rather than a full covariance matrix, we use sNES (Wierstra et al., 2011), which is a variation on NES that assumes independent variances between input elements.

Additionally, note that while this procedure fits perfectly for optimizing a variational quantum algorithm, it also works for PQC's that have parameters that aren't optimized for. All one needs to do is simply removing the $\theta$ update step for all fixed elements, so the parameter in question remains fixed and we only need to find an ideal $\sigma$ component for that element. This flexibility is utilized in our experiments outlined in Section 5.3. Furthermore, we note that our method can be applied to any parameterized quantum system, including those that utilize other error-mitigation methods. This allows us to easily boost robustness by combining with existing methods.

| Training Procedure | Deployed Model Use |
|---|---|
| **for** $N$ iterations **do** | |
|     **for** $k = 1 \dots \lambda$ **do** | |
|       $s_k \sim \mathcal{N}(0, I)$ | |
|       $z_k \leftarrow \theta + \sigma s_k$ | **Input :** $x, \theta$ |
|       $f_k \leftarrow (\Phi^{-1}(p_A) - \Phi^{-1}(p_B))/2$ | **for** $k = 1 \dots M$ **do** |
|     **end for** |     $s_k \sim \mathcal{N}(0, I)$ |
|     $s'_k \leftarrow$ Sort all $s_k$ w.r.t. $f_k$ |     $z_k \leftarrow \theta + \sigma s_k$ |
|     $u_k \leftarrow \frac{\max(0,\log(\lambda/2+1)-\log(k))}{\sum_{j=1}^{\lambda}\max(0,\log(\lambda/2+1)-\log(j))} - \frac{1}{\lambda}$ |     $c_{k,i} \leftarrow C(z_k, x)_i$ |
|     $\nabla_\theta J \leftarrow \sum_{k=1}^{\lambda} u_k s'_k$ | **end for** |
|     $\nabla_\sigma J \leftarrow \sum_{k=1}^{\lambda} u_k(s'^2_k - 1)$ | $p_i \leftarrow \frac{1}{M}\sum_{k=1}^{M} c_{k,i}$ |
|     $\theta \leftarrow \theta + \eta_\theta \cdot \sigma \cdot \nabla_\theta J$ | **Output :** $\arg\max_i p_i$ |
|     $\sigma \leftarrow \sigma \cdot \exp(\eta_\sigma/2 \cdot \nabla_\sigma J)$ | |
|     $\sigma \leftarrow \sigma + \eta_r \cdot \mathrm{reg}(\sigma)$ | |
| **end for** | |

Table 1: Training and use of our method outlined in Section 3. All variables are as defined by Theorem 3.1. See Section 3.3 for the possible definitions of the $reg(\cdot)$ function.

### 3.3 VARIANCE REGULARIZATION

After we have identified how to optimize to improve the right side of the bound in Theorem 3.1, we now turn to improve the left side in the bound. Notice that there is an implicit trade-off occurring in this bound; while the left side always benefits from a larger $\sigma$, making it larger introduces the risk of decreasing the accuracy of the smoothed classifier and in turn decreasing the right side of the bound. As such, to address this trade-off we add regularization into our optimization procedure. This allows us to define an optimization trade-off between improving the left-side of the bound via a large $\sigma$ and the right-hand side by encouraging high accuracy. We control this trade-off with a hyperparameter, and in practice we often sweep over many values of this coefficient to understand the nature of accuracy-robustness trade-off per experiment. To control the magnitude of $\sigma$, we utilize two types of regularization methods, both of which have similar performances in our experiments. (See Section B.2 for more details.)

## 4 METRICS

To test how robust an approach is under noise, we need to consider proper metric for quantification. While there are many ways to achieve this, understanding the bound from Theorem 3.1 from a geometric perspective is beneficial. Note that we can re-arrange the terms of the bound in Theorem 3.1 to form the equation of a hyper- ellipsoid:

$$\sum_{i=1}^{D} \frac{\delta_i^2}{(s_e \sigma_i)^2} < 1 \tag{1}$$

$$s_e = \frac{1}{2}\left(\Phi^{-1}(p_A) - \Phi^{-1}(p_B)\right). \tag{2}$$

In other words, any perturbation that exists within this hyper-ellipsoid will satisfy the bound and not cause a change in the model's prediction result.

Using this re-formulation, we can use the volume of this hyper-ellipsoid as a metric, as it is generally desirable to be robust to the largest possible space of perturbations. The volume of this hyper-ellipsoid is:

$$V = \frac{2\pi^{D/2}}{D\Gamma(D/2)} \prod_{i=1}^{D} s_e \sigma_i. \tag{3}$$

Using this geometric understanding, we can finally define a handful of useful metrics that capture a wide range of what most use-cases would desire to optimize for:

**Certified Area Geometric Mean** : The certified area (Equation 3) taken to the power of $\frac{1}{D}$, which is $V^{1/D}$. Since the certified area can be very small and vary wildly depending on the dimensionality of the problem, we use the geometric mean to make it easier to think about and compare from experiment to experiment. Conceptually this can be thought of as if we take the volume of the $D$-dimensional hyper-ellipsoid, re-shape it into a $D$-dimensional cube, and then calculate the length of the sides of the cube.

**Semi-Axis Average** : The average of the semi-axes of the certified hyper-ellipsoid (Equation 1), which is $\overline{s_e\sigma} = \frac{1}{D}\sum_{i=1}^{D} s_e\sigma_i$. Since conceptually each semi-axis length can be thought of as the maximum one can perturb a given parameter, this metric can be interpreted as the maximum that a parameter can change on average under noise.

**Semi-Axis Standard Deviation** : The standard deviation of the semi-axes of the certified hyper-ellipsoid, which is $\sqrt{\frac{1}{D}\sum_{i=1}^{D}(s_e\sigma_i - \overline{s_e\sigma})^2}$. Tracking this metric may provide insights into the sensitivity of the parameters in PQC. (See Section 4.2.)

**Smoothed Accuracy** : The accuracy of the smoothed classifier ($G_\sigma$ in Theorem 3.1), which is $\mathbb{E}_{(x,y)\sim D}[\mathbb{1}(G_\sigma(\theta, x) = y)]$. $G_\sigma(\theta, x)$ is calculated by sampling the underlying PQC with many different parameter-samples (using the Gaussian found by the process outlined in Section 3.2) and averaging the resulting probabilities.

In our experiments, all numeric are calculated using an average over a test dataset, none of which are seen during training. Before describing the experiments, we introduce the type of plots we will produce with the results.

## 4.1 ROBUSTNESS-ACCURACY TRADE-OFF

As mentioned in Section 3.3, there often exists a trade-off between accuracy and robustness when training smoothed classifiers. As such, we desire to understand the best trade-off we can achieve. In order to do this, we run a randomized hyperparameter sweep over the method described in Section 3. Specifically, we modulate over the regularization term strength term (Section 3.3) and all the hyperparameters of sNES. Looking at each level of smoothed accuracy, we select only the runs that achieve the best robustness metric for said accuracy and plot these points.

Note that this robustness-accuracy trade-off usually only exists for a small section of runs in the higher-accuracy regime, as any regularization strength coefficient that is too high will cause the accuracy margin of the model to dramatically decrease, which will in turn decrease the robustness metric. To properly understand the relevant frontier of this trade-off, we only plot the points on this frontier, and fit a line to them in order to understand the numerics a practitioner can expect to achieve on a similar problem without needing to do significant tuning. These plots can be seen in the first row of Figures 1, 2.

## 4.2 ROBUSTNESS-VARIANCE CORRELATION

We also produce plots to illustrate the relationship of robustness metrics (i.e. certified area geometric mean and semi-axis average) versus the semi-axis standard deviation. A high standard deviation indicates that the "robust space" hyper-ellipsoid (Equation 1) has some dimensions much longer than others, which indicates some parameters are more susceptible to noise than others. Conversely, a low standard deviation indicates that it is closer to a sphere, which indicates all parameters should tolerate near-equal amounts of noise.

For this analysis, we only include runs that achieve high accuracy, as these are the only points that are relevant to the "robustness-accuracy frontier" described in Section 4.1. We plot all runs that achieve above the minimum accuracy shown in the "robustness-accuracy frontier" plots. We then bin these runs to show the mean and standard deviation of the semi-axis standard deviation for all runs that achieve a similar robust metric. Similar to the robustness-accuracy trade-off plots, we also fit a line to these points to understand the overall corelation. These plots can be seen in the last rows of Figures 1, 2.

## 5 EXPERIMENTS

As randomized smoothing only works in classification tasks, we primarily consider phase classification tasks. We chose these tasks because they are important in condensed-matter physics, and as a result are often used in benchmarks (Carrasquilla & Melko, 2017; Broecker et al., 2017). In these problems, given the ground state quantum state, the objective is to predict what phase of the ground state originates from. For all of our experiments, we generate mutually exclusive train and test datasets of 50 samples each. The train dataset is used to train the model, and all statistics reported are averages over the entire test set. For all experiments, we do randomized hyperparameter sweeps in order to understand what we can optimally achieve, but in practice most hyperparameters (aside from the regularization strength term) worke well as long as they were within a reasonable range (See Section B.1 for more details).

### 5.1 CLASSIFICATION MODEL

Before describing the types of phase classification tasks we study, we first outline the PQC classifier we use to predict the phases. For all of our experiments, we use the Quantum Convolutional Neural Network (QCNN) (Cong et al., 2019). Specifically we use a form of QCNN that uses rotational and control X, Y, and Z gates to implement generic 1 and 2 qubit gates (Vatan & Williams, 2004; noa). We certify for all phase shift noise in any gate that uses a parameter-defined angle.

### 5.2 CLUSTER PHASE CLASSIFICATION

For this task, we consider the generalized cluster Hamiltonian, which is commonly studied in other works (Gil-Fuster et al., 2023; Caro et al., 2022). The Hamiltonian in this setup is:

$$H = \sum_{j=1}^{n} (Z_j + j_1 X_j X_{j+1} - j_2 X_{j-1} Z_j X_{j+1}).$$

This Hamiltonian contains 4 phases, depending on the values of the coefficients $j_1$ and $j_2$. The values that belong to each phase are illustrated in Figure 2 of Gil-Fuster et al. (2023) (included in our appendix as Figure 3 for convenience). For these experiments, the ground states are found exactly via diagonalization and loaded directly as the starting state of the circuit. Our data is uniformly sampled from $j_1 \in [-4, 4], j_2 \in [-4, 4]$. We present the results in Figure 1 with a Hamiltonian of 12 qubits.

**Results Discussion :** The first row of Figure 1 demonstrates the robustness-variance trade-off, where smoothed accuracy decreases with higher robustness metrics. Note that in these experiments, we are able to achieve a certified area geometric mean ranging from roughly $0.002$ to $0.018$ , and a semi-axis average ranging from $0.005$ to $0.045$ depending on the smoothed accuracy level. This means that one could likely expect to certify the robustness of similar experiments that experience this amount of phase-shift noise per parameter. While the usefulness of this level of robustness depends on each individual system, noise level, and desired accuracy, it is shown that it could be sufficient for certain systems (Bluvstein et al., 2022; Yi et al., 2024; Wood, 2020).

Furthermore, note that there is a clear correlation between high semi-axis standard deviation and our robustness metrics. This indicates that different parameters of the PQC have varying amounts of robustness to noise, and as a result we are able to leverage these differences to improve our overall robustness to noise. It illustrates the advantage of our approach with varying $\sigma$ for different parameters compared to a uniform robust radius, which provides more flexibility to be noise-resilient.

### 5.3 SPT QUANTUM STATE PREPARATION CLASSIFICATION

While in Section 5.2 we can certify against noise that occurs during the execution of our learned PQC, we do not study certification of error that occurs during state preparation since we simply directly load the ground state into the simulation. However, if we have a circuit which is used for preparing input data, which is common in a variety of applications (Cerezo et al., 2021), we can certify against noise that occurs in the parameterized gates of input circuit.

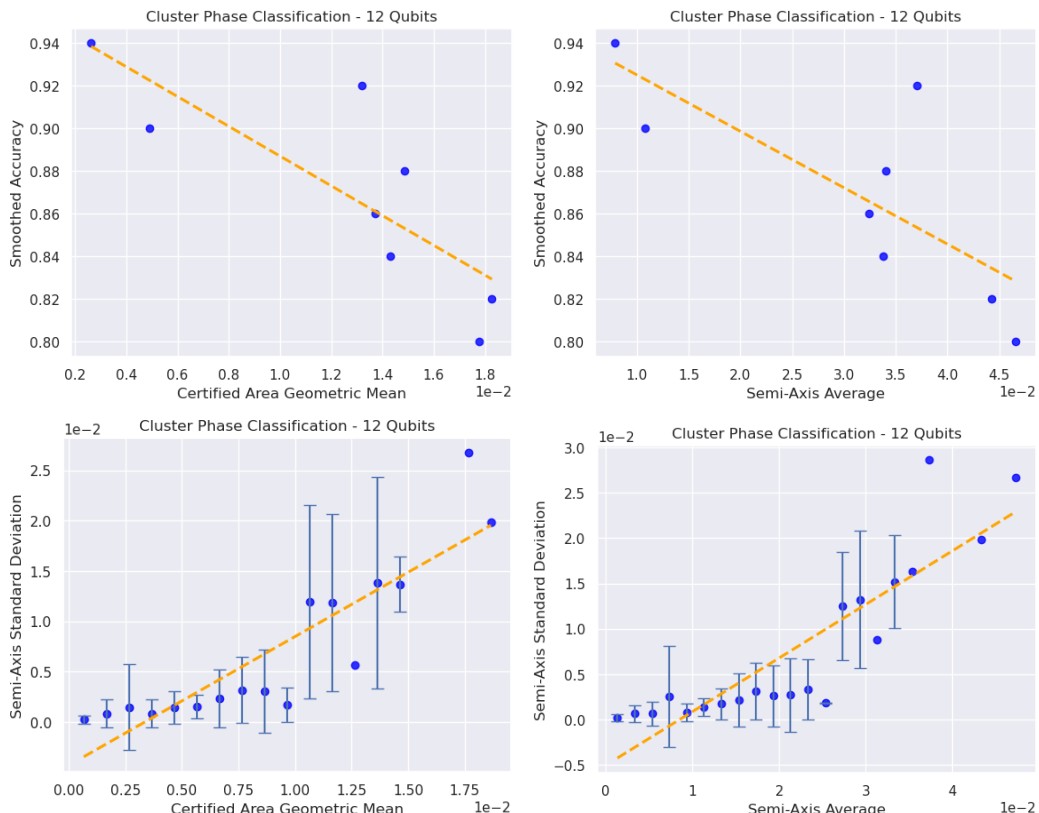

Figure 1: Phase classification for the generalized cluster Hamiltonian of 12 qubits, as outlined in Section 5.2. The first row illustrates the trade-off between accuracy and robustness, as described in Section 4.1. The last row shows the robustness-variance correlation, as described in Section 4.2. While our results may vary due to randomness and instability in optimization, we include a linear fit line to indicate the general trend.

To study this, we look at classification of phases proposed by Smith et al. (2022) to study a symmetry protected topological (SPT) phases of matter. Specifically we look at ground states for the Hamiltonian defined as

$$H = \sum_i [-g_{zz} Z_i Z_{i+1} - g_x X_i + g_{zxz} Z_i X_{i+1} Z_{i+2}]$$

$$g_{zz} = 2(1 - g^2), \; g_x = (1 + g)^2, \; g_{zxz} = (g - 1)^2$$

where $g \in [-1, 1]$ and the phase transition occurring at $g = 0$. The phase diagram is illustrated in Figure 1 of Smith et al. (2022) (included in our appendix as Figure 4). Using the circuit proposed by Smith et al. (2022), we can prepare the ground state, and then append a QCNN afterwards to learn to classify each phase of matter. We can then certify the robustness of this whole process using our proposed method. The results for these experiments include certification of all parameterized rotation X, Y, and Z gates in both the state preparation and the proceeding QCNN. Our data is sampled from $g \sim U(-1, 1)$. We present the results in Figure 2 with a Hamiltonian of 8 qubits.

**Results Discussion :** In these experiments, for the plots in the first row of Figure 2 we observe similar robustness-variance trade-off to those in Section 5.2. In this task, we are able to achieve a certified area geometric mean ranging from roughly 0 to 0.0175, and a semi-axis average ranging from 0 to 0.04 depending on the smoothed accuracy level. However, the second row of Figure 2 demonstrates a different robustness-variance correlation compared to the prior phase classification tasks. While there is still an overall trend between robustness and semi-axis standard deviation, the

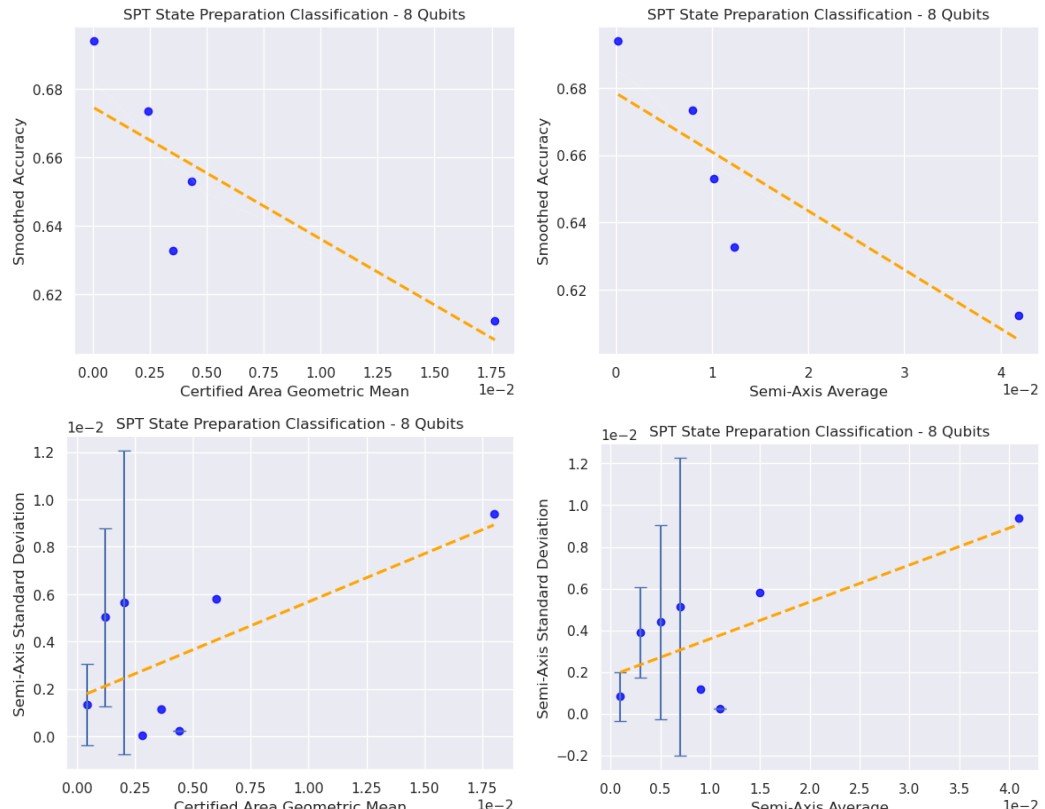

Figure 2: SPT quantum state preparation classification for the Hamiltonian of 8 qubits, as outlined in Section 5.3. The first row illustrates the trade-off between accuracy and robustness, as described in Section 4.1. The last row shows the robustness-variance correlation, as described in Section 4.2. While our results may vary due to randomness and instability in optimization, we include a linear fit line to indicate the general trend.

correlation is much weaker. Such differences may come from the fluctuation of optimization and the noise-resilient nature of the SPT state preparation task.

More specifically, this phenomenon could arise for two reasons. First, it may be a consequence of our optimization procedure. Since we can explicitly optimize only for accuracy, we must make certain decisions regarding the regularization of $\sigma$ to decrease the left-hand side of the bound in Theorem 3.1. While we make informed choices, we cannot directly optimize for the metrics we are tracking, relying instead on hyperparameter sweeps to identify optimal solutions. In the experiments described in Section 5.2, this strategy appears capable of producing a wide range of near-optimal results. However, in these experiments, producing optimal results seems more sporadic and less consistent. This interpretation suggests the need for further research to develop more targeted regularization techniques tailored to specific use cases.

Second, the SPT state preparation classification task may inherently exhibit a weaker correlation between semi-axis standard deviation and robustness metrics. If this is true, it implies that the parameters of the PQC in this task are more uniformly sensitive to noise compared to the previous task. Otherwise, the optimizer would likely have identified less-sensitive parameters and increased their semi-axes to enhance the robustness metric. If this hypothesis holds, it opens an intriguing avenue for future research to investigate which PQC tasks exhibit varying degrees of uniform robustness and to uncover deeper insights into why certain parameters are more sensitive to noise than others

## 5.4 LIMITATIONS

While we provide a provably noise-resilient approach for training PQC classier, there are limitations of the work. Since other types of noise observed in a quantum system might be difficult to translate back into the parameter-perturbation setting, it is an open question on how to generalize our results with provably guarantee to certify arbitrary types of quantum noise. Furthermore, due to resource constraint, we only provide numerical simulations and it will be valuable to perform experiments on real quantum devices in the future.

## 6 CONCLUSION

In this work we have developed a provably noise-resilient approach for training parameterized quantum circuit classifiers. Our method is flexible for any quantum circuit and easy to deploy on NISQ quantum device with a natural connection to a Evolutionary Strategies. This makes it extremely simple for practitioners to use our method on any of their existing experiments, both to enhance robustness and to gain insights into the sensitivity of the quantum devices. We have demonstrated our method's capabilities through experiments that not only certify the robustness of PQC classifiers, but also input parameterized quantum state-preparation. Future work could explore using the shape of the $\sigma$ vector to understand the sensitivity and importance of parameters in PQC classifiers, especially for different quantum ansatz. Additionally, further optimization and regularization of the PQC classifier could be customized based on specific performance metrics. Expanding the method to include a full-covariance matrix in randomized smoothing could also provide more flexibility in smoothing techniques. Our work integrates the frontier classical machine learning algorithm with quantum computation, opening up opportunities for robust applications of near-term quantum computers.

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

## A  ADDITIONAL EXPERIMENTAL DETAILS

In the first task of cluster phase classification, there are four phases as the following figure shows

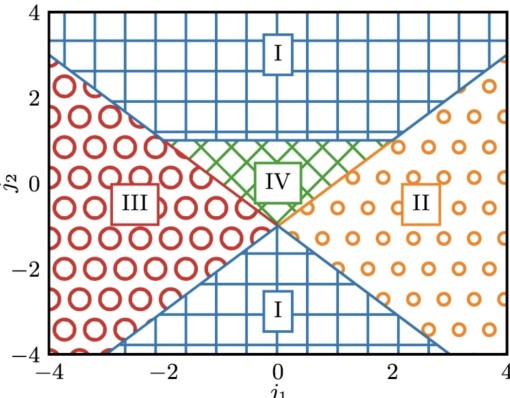

Figure 3: Figure 2 from Gil-Fuster et al. (2023). Illustrates the different phases of the generalized cluster phase-classification problem outlined in Section 5.2.

In the second task of SPT phase classification, there are two phases along the black line of $g$, as the following figure shows

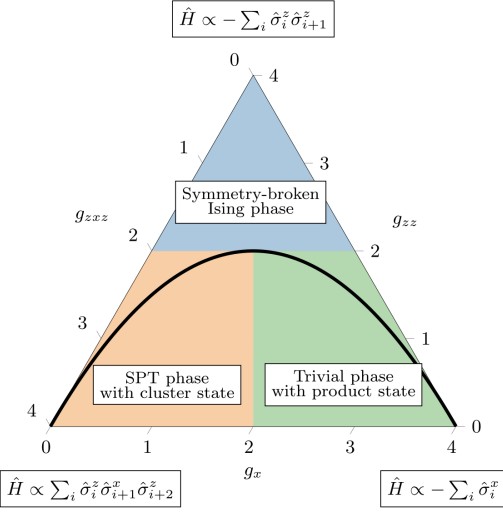

Figure 4: Figure 1 from Smith et al. (2022). Illustrates the different phases of the symmetry protected topological phase classification problem outlined in Section 5.3. The black line demonstrates the path of the $g$ variable as it is varied from $-1$ to $1$.

## B  IMPLEMENTATION DETAILS

### B.1  HYPERPARAMETERS

For all experiments we do randomized hyperparameter sweeps in order to understand what we can optimally achieve. Because we are looking at a combination of robustness and accuracy, it is difficult to definitively say which hyperparameters are optimal because it can vary from different levels of

accuracy and robustness. That being said, in our sweeps we found that the runs that produced optimal results (near-best robustness metrics for a given accuracy level) came from a wide range of hyperparameter values, and often a hyperparameter would only destroy performance if it was far too low or high. The acceptable range we found for each hyperparameter is as follows. $\sigma_0$ is the initial value of all elements of $\sigma$, and all other hyperparameter are as shown in table 1). The acronym (f.r.) indicates that this is the full range we tested, so the complete acceptable range may be larger.

|  | Generalized Cluster | SPT |
|---|---|---|
| $k$ | 10-40 (f.r.) | 10-30 (f.r.) |
| $\eta_\sigma$ | 1e-1 - 1e-3 (f.r.) | 1e-1 - 1e-3 (f.r.) |
| $\eta_\theta$ | 1 - 2e-2 | 1e-1 - 1e-4 (f.r.) |
| $\eta_r$ | 1e-2 - 1e-6 (f.r.) | 5e-2 - 1e-6 |
| $\sigma_0$ | 5e-1 - 1e-2 | 1e-1 - 1e-3 (f.r.) |

### B.2 REGULARIZATION

In this work we use two different types of regularization. The first is analogous to $L_2$ regularization in classical machine learning. Because this regularization is a common and simple yet effective choice for a wide variety of machine learning, it is appropriate in the absence of a more specific objective. Because each step of our ES optimization is intended to be similar to a gradient step in gradient descent, we will use the derivative of the $L_2$ norm (removing any constant terms, as these will be absorbed into the hyperparameter $\eta_r$ shown in table 1):

$$\text{reg}_1(\sigma) = c\nabla_\sigma \|\sigma\|_2^2 = \sigma$$

The second form of regularization is intended to specifically maximize certified area (see Equation 3). Similar to above, we will simply take the derivative of the certified area. However, because the derivative of this area directly is more complicated, we use the natural log of the area:

$$\text{reg}_2(\sigma) = c\nabla_\sigma \ln\left(\frac{2\pi^{D/2}}{D\Gamma(D/2)}\prod_{i=1}^{D} s_e\sigma_i\right) = c\nabla_\sigma \sum_{i=1}^{D}\ln(\sigma_i) = \sigma^{-1}$$

As to what type of regularization is best to use, it depends on the type of $\delta$ perturbations a practitioner expects to encounter. Despite the theoretical pros and cons of each approach, they seemed to perform comparably according to our metrics. We only saw significant differences between the two during relatively easy tasks we tried prior to the experiments presented in this paper where parameters could be perturbed extremely without affecting performance at all. In this case the L2 regularization would tend to increase the perturbations to extreme levels, whereas the area regularization would produce more moderate results.

### B.3 PROBABILITY ESTIMATION

When in practice using our method outlined in Section 3 and Theorem 3.1, it is impossible to exactly evaluate $p_A$ and $p_B$. This is because doing so would require you to compute the expectation over a Gaussian on the PQC. As such, when practically using this method, one must estimate the values of $p_A$ and $p_B$. To do so in such a way that still guarantees the conditions of theorem 3.1 hold, we can use some form of concentration inequality or confidence interval method. These mathematical methods will give us values $\underline{p_A}, \overline{p_B}$ such that $p_A \geq \underline{p_A}$ and $p_B \leq \overline{p_B}$ with probability $1 - \delta'$. Note that theorem 3.1 still holds if we replace $p_A$ with a lower bound on the true value, and likewise if we replace $p_B$ with an upper bound on the true value. As such, the whole theorem can still be used if these probabilities are estimated via sampling. And the more sampling a practitioner does, the better their estimate will become, and as a result also their robust certificate. In practice one can use whichever concentration inequality or confidence interval method they prefer, but in most cases the Clopper-Pearson confidence interval is used (Cohen et al., 2019; Tecot & Hsieh, 2021).

## C    THEOREM 3.1 PROOF

To show that $G_\sigma(\theta + \delta, x) = c_A$, it follows from the definition of $G_\sigma$ that we need to show that:

$$\mathbb{P}(\arg\max_{i \in \gamma} C(\theta + \delta + \epsilon, x)_i = c_a) \geq \max_{c_b \neq c_a} \mathbb{P}(\arg\max_{i \in \gamma} C(\theta + \delta + \epsilon, x)_i = c_b)$$

To show this, fix one class $c_b$ w.l.o.g. And for convenience we'll define the random variables:

$$T := \theta + \epsilon = (\theta, \Sigma)$$
$$Z := \theta + \delta + \epsilon = (\theta + \delta, \Sigma)$$

We will also define that:

$$\mathbb{P}(\arg\max_{i \in \gamma} C(T, x)_i = c_a) \geq \underline{p_A}$$
$$\mathbb{P}(\arg\max_{i \in \gamma} C(T, x)_i = c_b) \leq \overline{p_B}$$

Given this, our goal is to show that

$$\mathbb{P}(\arg\max_{i \in \gamma} C(Z, x)_i = c_a) > \mathbb{P}(\arg\max_{i \in \gamma} C(Z, x)_i = c_b)$$

To prove this, we define the half-spaces:

$$A := \{z : \lambda^\top (z - \theta) \leq \|\sigma \odot \lambda\| \Phi^{-1}(\underline{p_A})\}$$
$$B := \{z : \lambda^\top (z - \theta) \geq \|\sigma \odot \lambda\| \Phi^{-1}(1 - \overline{p_B})\}$$

Where $\lambda = \delta \oslash \sigma^{\circ 2}$, as per Lemma A.1 from Tecot & Hsieh (2021).

As in Section A.2 of Tecot & Hsieh (2021), it can be shown that $\mathbb{P}(T \in A) = \underline{p_A}$. Therefore, we know that $\mathbb{P}(\arg\max_{i \in \gamma} C(T, x)_i = c_a) \geq \mathbb{P}(T \in A)$. Hence we may apply Lemma A.1 from Tecot & Hsieh (2021) with $h := \mathbf{1}[\arg\max_{i \in \gamma} C(z, x)_i = c_a]$ to conclude:

$$\mathbb{P}(\arg\max_{i \in \gamma} C(Z, x)_i = c_a) \geq \mathbb{P}(Z \in A)$$

Similarly, algebra shows that $\mathbb{P}(T \in B) = \overline{p_B}$. Therefore, we know that $\mathbb{P}(\arg\max_{i \in \gamma} C(T, x)_i = c_b) \leq \mathbb{P}(T \in B)$. Hence we may apply Lemma A.1 from Tecot & Hsieh (2021) with $h := \mathbf{1}[\arg\max_{i \in \gamma} C(z, x)_i = c_b]$ to conclude:

$$\mathbb{P}(\arg\max_{i \in \gamma} C(Z, x)_i = c_b) \leq \mathbb{P}(Z \in B)$$

Now all we need to do is show that $\mathbb{P}(Z \in A) > \mathbb{P}(Z \in B)$, as this shows:

$$\mathbb{P}(\arg\max_{i \in \gamma} C(Z, x)_i = c_a) \geq \mathbb{P}(Z \in A) > \mathbb{P}(Z \in B) \geq \mathbb{P}(\arg\max_{i \in \gamma} C(Z, x)_i = c_b)$$

We can compute that (shown in the "Deferred Algebra" Section of Tecot & Hsieh (2021)):

$$\mathbb{P}(Z \in A) = \Phi\left(\Phi^{-1}(\underline{p_A}) - \frac{\langle \lambda, \delta \rangle}{\|\sigma \odot \lambda\|}\right)$$

$$\mathbb{P}(Z \in B) = \Phi\left(\Phi^{-1}(\overline{p_B}) + \frac{\langle \lambda, \delta \rangle}{\|\sigma \odot \lambda\|}\right)$$

Where $\langle \cdot, \cdot \rangle$ is the inner product of two vectors.

Via algebra, $\mathbb{P}(Z \in A) > \mathbb{P}(Z \in B)$ implies that $\|\delta \oslash \sigma\| < \frac{1}{2}(\Phi^{-1}(p_A) - \Phi^{-1}(p_B))$. Therefore $G_\sigma(\theta + \delta, x) = c_a$ for all $\delta$ vectors where this bound holds.

