# OpenReview forum: "Provably Noise-Resilient Training of Parameterized Quantum Circuits"
_ICLR.cc/2025/Conference — Submitted to ICLR 2025_

### Official Review · Reviewer_zX3k · 2024-10-31

**Soundness:** 3
**Presentation:** 2
**Contribution:** 1
**Rating:** 3
**Confidence:** 4

**Summary:**

This submission addresses the robustness of quantum machine learning models based on parametrized quantum circuits under perturbations of their parameters. The authors revisit the framework of certified robustness, and provide theory and an algorithm for training such parametrized circuits in a provably noise-resilient way. The authors phrase these results in the context of noisy quantum circuits, and explain that their framework can be easily used by practitioners, and that such noise-robustness guarantees can also provide insight in matters of interpretability.

**Strengths:**

Strengths:
Addresses an important issue: the negative effects of noise in parametrized quantum circuits.
Supports the claims in that indeed a training algorithm with robustness guarantees is provided.
Brings ideas from certified robustness to quantum machine learning, which had not been much looked at before.
Proposes a geometric picture to intuitively understand the core concepts of certified robustness.

The work is clearly presented, the analytic tools all look correct, and the numerical experiments are also clearly laid out.

**Weaknesses:**

I cannot recommend this submission be accepted for ICLR 2025 due to a lack of deeper and more significant contributions.

In particular:
1 The approach could be better motivated. The source of noise considered is one of the least detrimental ones in real quantum devices. Also, it is unclear how this paper is set apart from previous literature.

2 The theoretical contributions in this submission are limited to laying out existing framework, minimally adapting a well-known training algorithm, listing a few metrics, and proof-of-principle experiments.

3 The numerical experiments are not very convincing. First, the quantum systems studied are very small when compared to similar experiments in the literature in the last decade. Next, the trade-off between accuracy and robustness is barely recognizable in Figure 1, and completely absent in Figure 2, to the point that these plots are detrimental to the overall message of the paper. The rest of the plots (the comparison between area and semi-axis standard deviation) is very surprising, since when introducing the metrics, the authors already say “[the standard deviation] is likely not of particular note for any use…”. Following the geometric picture elaborated in Section 3.2., it should be clear that the area of an ellipsoid is closely related to the variance of its semi-axes. So, on the one hand, it is unclear why practitioners should be interested in this second analysis, and on the other hand the results follow almost directly from analytical geometry.

4 The choice of learning algorithm seems arbitrary. While it is true that evolutionary strategies have been used in the literature, it is unclear that they are preferable to gradient-based methods. The reference cited by the authors states that, in the worst-case, the complexity of evaluating gradients is linear in the complexity of evaluating the model times the number of parameters. Other references exist showing that this stops being true in specific cases of interest. And, even if it were true in general, evolutionary strategies are known to have much longer convergence times than gradient-based methods. The authors seem to ignore the fact that, even though computing each gradient iteration may take longer, this may be completely offset by the dramatically reduced number of iterations. This does not mean that focusing on evolutionary training algorithms is not well-justified, but the text is currently misleading in how uncommon an approach this is.

5 The authors mention that the proposed tool of certified robustness can also be used for parameters that are not optimized for. This makes a lot of sense. This further seems to highlight that this study is a convoluted spin of the simple message “take random perturbations of increasing size until the performance starts decreasing, independently for each parameter, and then report the size of the largest acceptable perturbation”. If this is accurate, this further highlights the lack of strength of the submission. If it is not accurate, it should be better clarified in the text.

**Questions:**

Questions and comments:
In the prior work section, the authors explain that the techniques of certified robustness and training of parametrized quantum circuits have been previously addressed and carry intrinsic interest. Still, as a reader, it seems that the presented techniques are a direct application of a previously-developed formalism. If this is not the case, the authors should explicitly delineate what are the novelties in Section 3.
The submission would be much more convincing if the authors had produced a general-case statement that further justifies why using this framework produces desirable results. Along the same lines, in Section 4.2.1. the authors open with “we desire to understand the best [robustness-accuracy] trade-off we can achieve”, and then proceed to perform limited numerical experiments, ignoring the question of best-possible trade-off.
The proposed approach is not compared to any other method. One would have expected to compare this “quantum-specific” tool to previously proposed “classical-general” tools applied to these quantum models.
The link to interpretability is rather weak. On the one hand, practitioners are most-likely aware of the relative relevance of each of their parameters. On the other hand, if the variance in the certified-robust parameter distribution is to be seriously considered as an interpretability method, then it should be properly benchmarked against other methods in the literature. Since the authors mention that this certified robustness can be linked to the sensitivity of the parameters, why did the authors not compare their results to standard sensitivity analysis, like using the Hessian or the Fisher information matrix?
In section 3.2. the volume of a high-dimensional ellipse is proposed as a metric for robustness. It is a well-known fact that the volume of a high-dimensional sphere vanishes asymptotically in the dimension of the ambient-space, so why would a metric based on this vanishing quantity be advantageous? This metric is introduced here, although later in the experiments different quantities are used, more closely-related to the volume of a high-dimensional prism than that of an ellipse. As a reader I found it very confusing why this metric was introduced, instead of the ones used in the experiments.
In section 4.4., the authors mention the potential users of their method, but they do not clarify who those users would be. This is again in relation to the choice of noise model. If the users ought to be researchers running these algorithms on hardware, the types of noise they are likely to encounter are not the ones captured in this study. If the users ought to be researchers performing simulations of quantum machine learning algorithms, then the inclusion of parameter noise becomes again less relevant. Who do the authors exactly envision as users of this method?

Feedback with the aim to improve the paper. These points are here to help, and not necessarily part of my decision assessment.
As a reader, several parts of the paper felt rushed and unpolished. The informal language in Sections 2. and 4.1. is distracting (subjective). There are a few grammar mistakes throughout the document. In Definition 3.2. the authors conclude with “for any noise perturbation delta from a certain distribution”, which makes no logical sense: it can either be “for all perturbations from a given domain”, or “with high probability over perturbations sampled from a given distribution”. The smoothed classifier as defined is a random variable, and it is never addressed whether or how to turn this random variable into a deterministic labeling function. In Section 3.3. the authors say “In general, what ES typically…”, which is logically incorrect: “in general” and “typically” are in contradiction. At the end of Section 3.4. the second regularization proposal ends with the inverse of sigma, where sigma is supposed to be a vector, so it is not clear what its inverse should be.
The authors invoke “noise resilience” in the title of the submission, referring to the uncontrollable effects that keep current hardware from being full-fledged quantum computers. Indeed, developing noise-resilient algorithms is an important task until we reach fault-tolerant quantum computers. That being said, the authors consider a single source of errors, and actually one of the least detrimental in real quantum devices: noise in the parameters. I would recommend the authors to either choose a more specific title, or to convincingly argue why parameter noise is important at present (notice that the goal of the field is to build noiseless quantum computers, so studying noise models must be justified from the point of view of real obstacles to that end).

---

> ### Author Response · Authors · 2024-12-03
>
> Thanks for the comments and the suggestions.
>
> Notes on Weaknesses Comments:
> 1. This paper is different from prior works in two ways. Firstly, we address parameter noise, which could be an important type of noise for certain experiments and is typically not considered in other mitigation schemes. This type of noise could happen in any quantum control setup where the experimental calibration is challenging and not accurate. Secondly, our method can be used on top of other error mitigation methods. So while our method itself may not directly address other forms of error, it can still be used in tandem with other methods to holistically address a wider range of errors.
> 2. We want to highlight that we make insightful connections that others have not made in the past. For instance, no other works that we are aware of apply randomized smoothing directly to the parameters (In all other works they certify non-trained data inputs) . Furthermore, our approach makes a novel connection between randomized smoothing and evolutionary strategies, in which ES naturally provides a resource efficient robust training method for quantum machine learning models. This has the additional benefit of not requiring gradient computation, which can be costly in many quantum settings. In addition, our approach is flexible and can be used for both input noise on quantum states and noise in quantum processes.
> 3. We do acknowledge that the original Figure 2 experiments produced strange-looking results and were not well explained. In the updated paper, we replace this with a new experiment that is more systematic and significant in physics. The new results do actually show a robustness-accuracy tradeoff similar to the nature of the first figure. Furthermore, we update the plots and discussions around variance to make our points clearer. Specifically we highlight that there is a correlation between high semi-axis standard deviation and our robustness metrics, which illustrates the advantage and flexibility of our approach with varying σ for different parameters compared to a uniform robust radius. With respect to the size of our experiments, we have demonstrated our approach successfully up to 12 qubits in the current work, which is a pretty good size in simulations. It could be an interesting future work if one is able to get access to real large scale quantum computers.
> 4. While gradient based methods may converge faster, they often have a cost-per-iteration that negates the benefits of this faster convergence. In practice, it is an open question for large scale quantum computers whether gradient or evolutionary strategies are better. However, note that while our work uses ES because it fits naturally into the formulation, it is still easily possible to optimize our objective using gradient-based methods. In this work it is encouraging that even with ES we have found good performances, so it is likely we would also find good performance in settings where using gradient descent makes more sense.
> 5. This statement isn’t fully correct, because there are a number of considerations that go into how much we perturb each parameter (see our section on regularization). But we would like to point out that this is simply a small extension that can be made to include fixed parameters as part of the robust certificate, which can allow us to certify noise on input quantum states. The majority of our work relates to how one can train and certify parameters at once while requiring no additional work or computation to existing methods.
>
> Answers to questions are included in additional official comments below.
>
> We thank the reviewer again for the comments and the suggestions to improve our work. With our discussion and the updated version of the paper including new experiments and elaboration, it will be appreciated that the reviewer can consider raising the score.

---

> ### Author Response · Authors · 2024-12-03
>
> **Question:** “In the prior work section, the authors explain that the techniques of certified robustness and training of parametrized quantum circuits have been previously addressed and carry intrinsic interest. Still, as a reader, it seems that the presented techniques are a direct application of a previously-developed formalism. If this is not the case, the authors should explicitly delineate what are the novelties in Section 3. The submission would be much more convincing if the authors had produced a general-case statement that further justifies why using this framework produces desirable results. Along the same lines, in Section 4.2.1. the authors open with “we desire to understand the best [robustness-accuracy] trade-off we can achieve”, and then proceed to perform limited numerical experiments, ignoring the question of best-possible trade-off. The proposed approach is not compared to any other method. One would have expected to compare this “quantum-specific” tool to previously proposed “classical-general” tools applied to these quantum models. “
>
> **Answer:** We want to highlight that we make insightful connections that others have not made in the past. For instance, no other works that we are aware of apply randomized smoothing directly to the parameters (In all other works they certify non-trained data inputs) . Furthermore, our approach makes a novel connection between randomized smoothing and evolutionary strategies, in which ES naturally provides a resource efficient robust training method for quantum machine learning models. This has the additional benefit of not requiring gradient computation, which can be costly in many quantum settings. In addition, our approach is flexible and can be used for both input noise on quantum states and noise in quantum processes. The top row of each figure is intended to demonstrate the robustness-accuracy tradeoff, as they are the best runs we achieved at each level of accuracy.  As to comparing against other methods, this is complicated in this setting. This is because we focus on parameter noise, whereas the prior works that we saw focused on mid-circuit errors. So this makes it difficult to do a direct translation between the two without some major assumptions, which can often defeat the purpose of doing such a comparison in the first place. In our eyes we see this as a method used for a fundamentally different purpose, and one that is easy and convenient to use in most settings. But you bring up a good point that perhaps we could find some illustrative specific examples that might give useful insight.
>
> **Question:** “The link to interpretability is rather weak. On the one hand, practitioners are most-likely aware of the relative relevance of each of their parameters. On the other hand, if the variance in the certified-robust parameter distribution is to be seriously considered as an interpretability method, then it should be properly benchmarked against other methods in the literature. Since the authors mention that this certified robustness can be linked to the sensitivity of the parameters, why did the authors not compare their results to standard sensitivity analysis, like using the Hessian or the Fisher information matrix?”
>
> **Answer:** We update the plots and discussions around variance to make our points clearer. We make a more precise discussion on robustness-variance correlation instead of interpretability. Specifically we highlight that there is a correlation between high semi-axis standard deviation and our robustness metrics, which illustrates the advantage and flexibility of our approach with varying σ for different parameters compared to a uniform robust radius. While it is not clear how the sensitivity of Hessian or Fisher information matrix is exactly connected to the sensitivity in robust certification, it will be an interesting research question for systematic study in the future.

---

> ### Author Response · Authors · 2024-12-03
>
> **Question:** “In section 3.2. the volume of a high-dimensional ellipse is proposed as a metric for robustness. It is a well-known fact that the volume of a high-dimensional sphere vanishes asymptotically in the dimension of the ambient-space, so why would a metric based on this vanishing quantity be advantageous? This metric is introduced here, although later in the experiments different quantities are used, more closely-related to the volume of a high-dimensional prism than that of an ellipse. As a reader I found it very confusing why this metric was introduced, instead of the ones used in the experiments.”
>
> **Answer:** With respect to certified volume, it is true that it vanishes in high-dimension, but it is meaningful for comparison when put next to experiments with the same number of parameters (And we also use the geometric mean to help diminish this vanishing effect for comparing experiments of different dimensions). We also include the metric of semi-axes average, which does not vanish as dimension increases and can be used for comparing experiments of different dimensions without this vanishing effect. In general, the two metrics are complementary and provide different information in understanding the robustness of the end result.
>
> **Question:** “In section 4.4., the authors mention the potential users of their method, but they do not clarify who those users would be. This is again in relation to the choice of noise model. If the users ought to be researchers running these algorithms on hardware, the types of noise they are likely to encounter are not the ones captured in this study. If the users ought to be researchers performing simulations of quantum machine learning algorithms, then the inclusion of parameter noise becomes again less relevant. Who do the authors exactly envision as users of this method?”
>
> **Answer:** We envision our users to be primarily experimentalists running algorithms on hardware. In this setting, parameter noise can be significant and could happen in any quantum control setup where the experimental calibration is challenging and not accurate. We provide a way to address this noise whereas other works typically do not consider it. Furthermore, this method can be used on top of other error mitigation methods. So while this method itself may not directly address other forms of error, it can still be used in tandem with other methods to holistically address a wider range of errors.
>
> **Question:** “As a reader, several parts of the paper felt rushed and unpolished. The informal language in Sections 2. and 4.1. is distracting (subjective). There are a few grammar mistakes throughout the document. In Definition 3.2. the authors conclude with “for any noise perturbation delta from a certain distribution”, which makes no logical sense: it can either be “for all perturbations from a given domain”, or “with high probability over perturbations sampled from a given distribution”. The smoothed classifier as defined is a random variable, and it is never addressed whether or how to turn this random variable into a deterministic labeling function. In Section 3.3. the authors say “In general, what ES typically…”, which is logically incorrect: “in general” and “typically” are in contradiction. At the end of Section 3.4. the second regularization proposal ends with the inverse of sigma, where sigma is supposed to be a vector, so it is not clear what its inverse should be. ”
>
> **Answer:** We thank the reviewer for this feedback, and in our revision of the paper have fixed many of these issues. We re-structured the flow of the paper and polished the grammar significantly. Even though it is not shown in the revision we have submitted here, we have updated Definition 3.2 to use the wording suggested here, and we have corrected Definition 3.3 to make the smoothed classifier a deterministic labeling function.

---

> ### Author Response · Authors · 2024-12-03
>
> **Question:** “The authors invoke “noise resilience” in the title of the submission, referring to the uncontrollable effects that keep current hardware from being full-fledged quantum computers. Indeed, developing noise-resilient algorithms is an important task until we reach fault-tolerant quantum computers. That being said, the authors consider a single source of errors, and actually one of the least detrimental in real quantum devices: noise in the parameters. I would recommend the authors to either choose a more specific title, or to convincingly argue why parameter noise is important at present (notice that the goal of the field is to build noiseless quantum computers, so studying noise models must be justified from the point of view of real obstacles to that end).”
>
> **Answer:** We would be happy to change the title to “Provably Parameter Noise-Resilient Training of Parameterized Quantum Circuits”. Meanwhile, we would like to highlight that this type of parameter noise could be important. It happens in any quantum control setup where the experimental calibration is challenging and not accurate, so it is not necessarily an insignificant source of noise. Furthermore, our method can be used on top of other error mitigation methods. So while our method itself may not directly address other forms of error, it can still be used in tandem with other methods to holistically address a wider range of errors.

---

### Official Review · Reviewer_hAo8 · 2024-10-31

**Soundness:** 2
**Presentation:** 2
**Contribution:** 2
**Rating:** 3
**Confidence:** 4

**Summary:**

The paper proposes a noise-resilient training theory and algorithm for parameterized quantum circuit (PQC) classifiers. The method guarantees resilience to parameter noise by integrating randomized smoothing techniques from classical machine learning. The authors establish a connection to evolutionary strategies for optimization and demonstrate the effectiveness of their approach on phase classification tasks. The paper also highlights the potential for interpretability insights into quantum models.

**Strengths:**

- For the specific task of parameterized quantum circuit classifiers (the model considered is indeed not the most general PQC classifier), this work makes an interesting attempt to develop a noise-resilient training theory.
- The paper is well written.
- This work makes attempts to show detailed experiments on some tasks of practical interest.

**Weaknesses:**

- The paper does not provide a clear and compelling justification for focusing on the PQC classifiers introduced in this paper, in particular given the current limitations in their trainability due to barren plateaus. The authors should address this concern and explain why their noise-resilient training approach is still relevant and important in light of these limitations.
- The PQC classifiers considered in this paper are somehow limited, and it is not clear how they are useful for practice. It would be better to show that the PQC classifier formulation would be particularly useful, which would help clarify the real-world relevance of their work.
- The proposed method does not take into account the practical issue of measurement accuracy in quantum systems. Measurement errors can significantly impact the performance of quantum algorithms.
- The numerical experiments presented in the paper may not be sufficiently convincing, as recent research (arXiv:2408.12739) has shown that Quantum Convolutional Neural Networks, which are used in the experiments, are effectively classically simulable. The authors should address this concern.
- The theoretical analysis in the paper appears to be mainly based on techniques directly adapted from classical methods. As a result, the novelty of the theoretical contributions may be limited, and the results may not be particularly surprising in the context of quantum machine learning.

**Questions:**

- Can the authors explicitly address how their approach relates to or mitigates the barren plateau problem? It would be helpful to understand if the proposed method has any inherent advantages in avoiding or reducing the impact of barren plateaus.
- Could the authors discuss the potential advantages of their Parameterized Quantum Circuit (PQC) classifier formulation compared to more general models?
- Can the authors provide more insights into why the PQC classifier formulation considered in this work is important and practically useful? Elaborating on the practical significance and potential applications of the proposed approach would strengthen the contribution of the paper.
- How might the proposed method be extended to account for measurement errors? Addressing the robustness of the method to measurement errors would enhance the completeness of the work.
- Can the authors explicitly discuss the implications of the cited classical methods in relation to their work? It would be valuable to understand how the proposed method might apply to other, non-classically simulable quantum models, or to explain why the approach is still valuable even for classically simulable models. This discussion would help contextualize the significance of the contribution within the broader landscape of quantum and classical methods.

---

> ### Author Response · Authors · 2024-12-03
>
> Thanks for the comments and the suggestions.
>
> Notes on Weaknesses Comments:
> 1. Barren plateaus is an important but orthogonal topic under our current framework. There are more understanding and solutions for barren plateaus nowaday in the community, such as proper architecture and initialization of quantum circuits.
> 2. PQC is one of widely used quantum machine learning models that can be realized in almost each type of quantum hardwares. It has nice features and advantages discussed in the citations we mention in the introduction. We also have more discussion later on this point when we answer the questions.
> 3. Measurement errors are included in the certificate of this work, as we simply need p_a and p_b in the theorem to calculate the robustness bound. While measurement errors can certainly make these probability numbers worse than their true value, they are still certified against by our theorem. We also have more discussion later on this point when we answer the questions.
> 4. We want to point out that our method is applicable to quantum circuits that are both classically simulable or beyond classical simulation. The goal of our work is to provide additional robustness for quantum experimentalists, no matter what circuit they are attempting to run. For simulation of small systems, all quantum circuits are classically simulable and hence we just choose QCNN. We would be happy to apply our approach to large-scale quantum computers if they are available for access. We also have more discussion later on this point when we answer the questions.
> 5. While our theory is also applicable to classical models, it is the first time that we generalize the theory to quantum setup. There are a few nice features of our approach in the quantum setting.  For instance, no other works that we are aware of apply randomized smoothing directly to the parameters (In all other works they certify non-trained data inputs) . Furthermore, our approach makes a novel connection between randomized smoothing and evolutionary strategies, in which ES naturally provides a resource efficient robust training method for quantum machine learning models. This has the additional benefit of not requiring gradient computation, which can be costly in many quantum settings. In addition, our approach is flexible and can be used for both input noise on quantum states and noise in quantum processes.
>
> Here are the answers to Questions:
> 1. In this work, we aim to provide a convenient and efficient addition to evolutionary strategies that allows an experimentalist to gain robustness guarantees for nearly free. The discussion of barren plateaus is a separate research topic, depending on the tasks  and the initialization of quantum circuits. The issue of barren plateaus could be avoided with proper design of the quantum circuits. If we work with such a design under our framework, then our method is also applicable.
> 2. We simply use PQCs and phase classification as an example because it is close to what an experimentalist might use, while also being general enough to apply to multiple types of quantum systems (we have listed the relevant citation in the introduction which discusses the advantages and the applications of PQC). Meanwhile, our approach is generic and flexible to apply to other types of models.
> 3. Related to the above point, PQCs are important because they can be realized in almost every type of quantum hardwares. All the more specific models we know run the risk of being too focused on only one type of setup. We have listed the relevant citation in the introduction which discusses the advantages and the applications of PQCs.
> 4. Measurement errors are included in the certificate of this work, as we simply need p_a and p_b in the theorem to calculate the robustness bound. While measurement errors can certainly make these probability numbers worse than their true value, they are still certified against by our theorem.
> 5. Our approach provides a general tool to be used for robustness guarantees for quantum systems, irrespective of whether they are classically simulatable or not. In our case we test on classical simulations because this is all that we have ready access to. It is a very good suggestion and will be significant to apply our method to large-scale quantum devices beyond classical simulation.
>
> We thank the reviewer again for the comments and the suggestions to improve our work. With our discussion and the updated version of the paper including new experiments and elaboration, it will be appreciated that the reviewer can consider raising the score.

---

### Official Review · Reviewer_179w · 2024-11-03

**Soundness:** 2
**Presentation:** 2
**Contribution:** 2
**Rating:** 3
**Confidence:** 5

**Summary:**

This paper applied the smoothing technique in certified robustness of classical machine learning to a quantum parametrized classifier model. The results imply that simply adding a careful designed noise parameter to the parameter optimization can enhance and certify the robustness of a variational quantum algorithm. However, I find some major issues regarding both the theoretical and numerical results:

**Strengths:**

The paper offers a plug-in approach to certify robustness of a variational quantum algorithm.

**Weaknesses:**

The assumption made for robustness certification is not very practical.
The numerics results are relatively weak.

**Questions:**

1. How does the framework extend to the phase parameters in a quantum gate?
2. Theorem 3.1 states the certification of robustness with a high probability under the assumption of noise. I did not find the discussion about this ``high probability” in the manuscript.
3. How does the shot limitation impact the certification? A theorem/lemma regarding to the statistic noise would be beneficial.
4. It makes sense to indicate a more sensitive parameter under this scheme. However, the word  “interpretability” in Section 4.2.2 looks over claiming as it simply states about the “robustness” and looks like a repartition of the first point “shape” in the same subsection.
5. The numerical results are relatively weak. Many experiment details are missing. What is PQC model used in the experiment? What is size of the PQC parameters? How many training data used? What is the additional computation cost when using the certified approach?
6. In the phase classical problem, why to choose the initial state as the ground state?
7. As the authors agreed in the discussion of the limitation, the assumption of treating quantum noise as parameter perturbations is not realistic. Some potential applications will be in the quantum control rather than VQA or quantum machine learning.

---

> ### Author Response · Authors · 2024-12-03
>
> Thanks for the comments and the suggestions.
>
> Notes on Weaknesses Comments:
> 1. Thanks for the comments. We have included new experiments with strong numeric support in the updated version to demonstrate our approach. The assumption we have in the paper is on the noise for parameters, which is physically relevant in all kinds of experiments. We have further elaborated that such scenarios can happen during the information processing of a quantum circuit or for an input quantum state, which are demonstrated in the phase classification and the quantum state preparation tasks.
>
> Here are the answers to Questions:
> 1. Our framework can be directly used for phase parameters. In our experiments, we have certifying robustness against changes in the parameter of a quantum gate, which are the phase parameters.
> 2. The theorem itself holds as-is without any caveats like high probability. However, the catch is that p_a and p_b can’t be computed directly because they require computing the expectation of the parameter gaussian over our quantum circuit. So in practice, we compute p_a and p_b via sampling, and then find upper and lower bounds on the true values of p_a and p_b (with high probability) using confidence intervals. As to why we left out these details in the paper, we felt that explaining all of this simply added unnecessary confusion to our target audience (quantum experimentalists) who are likely not familiar with these machine learning methods. And in practice these details make little difference to the end result (You just decide how much you can sample your circuit, and then your end result is a relatively constant factor away from the mean of your samples). Thanks for raising a good point that, while we may gloss over this in the main text to avoid confusion, it is an important detail to address in the appendix for practitioners looking to apply our method on a real device. So we added section B.3 in the appendix to include these details.
> 3. Measurement noises are included in the certificate of this work, as we simply need p_a and p_b in the theorem to calculate the robustness bound. While measurement errors can certainly make these probability numbers worse than their true value, they are still certified against by our theorem.
> 4. We have updated the paper to clarify the wordings as it is suggested. In this paper we just try to present this point from a numerical perspective via standard deviation, and point out that high standard deviation could have a correlation with robustness metric. This is particularly clear in the first experiment in the bottom row shown in Figure 1.
> 5. We have updated the main text and the appendix to include the details of the experiments. The classification model we use is explained and referenced in section 5.1 (QCNN). QCNNs have a fixed structure depending on the number of qubits, so their size is directly proportional to the number of qubits used in each experiment. The amount of training data used is explained at the top of section 5. There is no additional computational cost over the basic evolutionary strategies, both when training the quantum circuit and when using it to predict classes on new data. The only thing that may cost more is a practitioner may decide to sample more than they need to to boost their confidence in the prediction margin they compute (related to the answers for question 2 & 3). We added this point to the paper at the end of Section 3.1.
> 6. The purpose of a phase classification task is to predict which phase of the Hamiltonian produced the given ground state. Hence, it is standard and proper to use different ground states as initial states and see whether the QCNN can classify those states with the corresponding properties.
> 7. Thanks for the comments and it is a good point that our approach is also situated for quantum control. Indeed, in all realistic experimental setup for QAOA and VQE, one needs to use quantum control methods to choose some parameters for the quantum device, where the noise we discuss will come into play. The purpose of this work is to be a simple and very low-cost way of addressing this type of noise. However, note that while this method doesn’t explicitly address mid-circuit noise, the theorem still holds even in the presence of it. So this method can be put on-top of whatever noise-mitigation method you like to add some additional robustness. We add this information to the paper in our contribution point #2 in the introduction.
>
> We thank the reviewer again for the comments and the suggestions to improve our work. With our discussion and the updated version of the paper including new experiments and elaboration, it will be appreciated that the reviewer can consider raising the score.

---

### Official Review · Reviewer_Q8ZY · 2024-11-04

**Soundness:** 3
**Presentation:** 2
**Contribution:** 2
**Rating:** 3
**Confidence:** 3

**Summary:**

The authors develop a training method that guarantees resilience to parameter noise in quantum circuits for parameterized quantum circuit classifier. The authors give the theoretical guarantees and demonstrate with two experiments. The proposed method is closely linked to Evolutionary Strategies, which are a class of gradient-free optimizer. The framework also emphasizes interpretability by allowing researchers to analyze which parameters contribute most significantly to robustness after the optimization process.

**Strengths:**

1. The authors creatively combine concepts from classical robustness theory with quantum computing, offering a new perspective on how to handle parameter noise.

2. The authors give the theoretical foundation of the work.

3. The authors make a connection to Evolutionary Strategies, which makes achieve robustness certificates without the need for calculating expensive quantum gradients.

**Weaknesses:**

1. The presentation of the experiments makes it difficult to understand; the discussion does not refer to the experimental results but merely provides statements. It would be beneficial to add one or two paragraphs to discuss the experimental results. The connection between the discussions and the experimental results is not clear regarding variance-induced improvement and variance-indicated interpretability.

2. The authors conduct experiments primarily focused on phase classification and state preparation tasks. However, these may not fully represent the diverse range of applications of parameterized quantum circuits (PQC).

3. There is a lack of comparison to other methods. This can be improved by including experiments and discussion with quantum certified robustness methods as discussed in prior work, or even with standard quantum experiments using noise mitigation methods.

4. There is a lack of detailed descriptions of the experimental setup and hyperparameter sensitivity analysis. As one example, the authors mention two different variance regularizations but do not conduct any experiments to evaluate the effectiveness of these regularizations. This omission leaves readers uncertain about the impact and optimal choice of these regularizations."

5. Although the paper claims that their method provides insights into which parameters have larger robustness guarantees, it does not elaborate on how these insights can be practically utilized or interpreted by users. It would be beneficial to request specific examples from the authors that illustrate how these insights could be applied in practice. This would encourage them to provide more concrete information regarding the practical value of their method's interpretability.

6. Table 1 is hard to understand at first look, it might need better presentation.

**Questions:**

1. How does your method compare to prior work and the models with noise mitigation strategies in terms of performance metrics such as accuracy, robustness, and computational efficiency? Can you provide benchmarks against other state-of-the-art methods?

2. What is the sensitivity of your results to the choice of hyperparameters?

3. As circuit complexity increases, what are your thoughts on the scalability of your training method? Have you explored strategies to maintain efficiency with larger datasets or more complex quantum circuits?

---

> ### Author Response · Authors · 2024-12-03
>
> Thanks for the comments and the suggestions. Here are the answers to Questions:
>
> 1. We have provided a new version with better presentation on the algorithms and the results.
> 2. In this work, we have established the theorem for classification tasks so that we focus on phase classification and quantum state preparation tasks, which are important representatives. We will leave other tasks beyond this framework for future work.
> 3. We first want to point out that our approach can be used on top of other error mitigation techniques, which is one of the features of our approach. Meanwhile, providing benchmarks against prior works is tricky in our setting. This is because we focus on parameter noise, whereas the prior works that we saw focused on mid-circuit errors. So this makes it difficult to do a direct translation between the two without some major assumptions, which can often defeat the purpose of doing such a comparison in the first place. From this perspective, our method is used for a fundamentally different purpose, and one that is easy and convenient to use in most settings. Meanwhile, it is a good point as you mention perhaps we could find some illustrative specific examples that might give useful insight. We would be interested in hearing your suggestions for any such examples we could use.
> 4. In this paper we sweep over all hyperparameters of our method (section 3.3, 4.1) in order for us to understand the optimal robustness-accuracy tradeoff of our method. So sensitivity is a hard thing to define because it can change depending where on the  robustness-accuracy frontier you are. That being said, to give a broad generalization, we noticed that our method is pretty robust to hyperparameter changes. In our sweeps, most of our hyperparameters produced good results as long as they remained within reasonable ranges (which often spanned ~3 orders of magnitude). We added section B.1 in the appendix to explain more details on this topic.
> 5. We have updated the paper and the experiment to discuss how our approach can provide further insight into the sensitivity of the parameters. In particular, our approach allows different robust perturbation for different parameters, which provides more flexibility based on the sensitivity of each parameter. This can be explicitly seen in the last row of Figure 1, where there could be a strong robustness-variance correlation for certain tasks. In those cases, we find that a high variance of parameter perturbation is correlated with larger robustness, which indicates the variance of the parameter sensitivity could be beneficial.
> 6. We have updated Table I with more information to be clear.
>
> Notes on Weaknesses Comments:
> 1. We want to mention that our approach can be used on top of other error or noise mitigation techniques to gain further improvement. From this perspective, we do not focus on comparison with other noise mitigation methods.
> 2. As it relates to the regularization strategies, similar to the hyperparameters we elected to not to provide analysis on the differences because in many cases they produced similar results. We added section B.1 in the appendix to explain more details on this topic for readers that wish to know.
> 3. Since we have established a nice connection with noise robust training and evolutionary strategies, our method is quite scalable in terms of computational resources to large systems by avoiding the complexity of computing quantum gradients. Indeed, our cost is the same as the standard training method. We have demonstrated our approach successfully up to 12 qubits in the current work, which is a pretty good size in simulations. In terms of performance, it may depend on the nature of the task and the types of the evolutionary algorithms, which could be an interesting future work if one is able to get access to real large scale quantum computers.
>
> We thank the reviewer again for the comments and the suggestions to improve our work. With our discussion and the updated version of the paper including new experiments and elaboration, it will be appreciated that the reviewer can consider raising the score.

---

### Author Response · Authors · 2024-12-03

We thank the reviewer again for the comments and the suggestions to improve our work. We have updated our paper to include an important new quantum state preparation experiment, which provides much clearer and insightful results than the previous experiments. Furthermore, we have also made a number of modifications to improve the overall clarity of our paper. This includes updating the robustness-variance correlation plots, changing the overall organization and flow of our paper, and adding a number of sections to the appendix to address finer details. We feel that these updates greatly improve the paper’s ability to highlight both the usefulness of our method and the insights we are able to learn from it. We hope that our updates address the reviewers’ concerns and it will be appreciated that the reviewer can consider raising the score.

---

### Meta-Review · Area_Chair_nuYM · 2024-12-25

**Metareview:**

The authors propose a noise-resilient training procedure for parametrized quantum circuits. The techniques are related to randomized smoothing from classical ML. While there are some experimental results, the general consensus among the reviewers is that substantial additional work, especially in comparison to other methods is needed for this work to be suitable for publication.

**Additional Comments On Reviewer Discussion:**

The reviewers engaged well with the authors during the rebuttal phase.

---

### Decision · Program_Chairs · 2025-01-22

Reject